# Waste *Citrus reticulata* Assisted Preparation of Cobalt Oxide Nanoparticles for Supercapacitors

**DOI:** 10.3390/nano12234119

**Published:** 2022-11-22

**Authors:** Rishabh Srivastava, Shiva Bhardwaj, Anuj Kumar, Rahul Singhal, Jules Scanley, Christine C. Broadbridge, Ram K. Gupta

**Affiliations:** 1Department of Physics, Pittsburg State University, Pittsburg, KS 66762, USA; 2National Institute of Material Advancement, Pittsburg, KS 66762, USA; 3Nano-Technology Research Laboratory, Department of Chemistry, GLA University, Mathura 281406, Uttar Pradesh, India; 4Department of Physics and Engineering Physics, Central Connecticut State University, New Britain, CT 06050, USA; 5Connecticut State Colleges and Universities (CSCU) Center for Nanotechnology, Southern Connecticut State University, New Haven, CT 06515, USA; 6Department of Chemistry, Pittsburg State University, Pittsburg, KS 66762, USA

**Keywords:** Co_3_O_4_ nanoparticles, characterization, supercapacitors, charge storage, *Citrus reticulata*

## Abstract

The green, sustainable, and inexpensive creation of novel materials, primarily nanoparticles, with effective energy-storing properties, is key to addressing both the rising demand for energy storage and the mounting environmental concerns throughout the world. Here, an orange peel extract is used to make cobalt oxide nanoparticles from cobalt nitrate hexahydrate. The orange peel extract has *Citrus reticulata*, which is a key biological component that acts as a ligand and a reducing agent during the formation of nanoparticles. Additionally, the same nanoparticles were also obtained from various precursors for phase and electrochemical behavior comparisons. The prepared Co-nanoparticles were also sulfurized and phosphorized to enhance the electrochemical properties. The synthesized samples were characterized using scanning electron microscopic and X-ray diffraction techniques. The cobalt oxide nanoparticle showed a specific capacitance of 90 F/g at 1 A/g, whereas the cobalt sulfide and phosphide samples delivered an improved specific capacitance of 98 F/g and 185 F/g at 1 A/g. The phosphide-based nanoparticles offer more than 85% capacitance retention after 5000 cycles. This study offers a green strategy to prepare nanostructured materials for energy applications.

## 1. Introduction

The demand for energy leads to the continuous depletion of fossil fuels. Today, due to the requirement for a large amount of energy, scientists tend to move toward the storage application with the help of energy storage devices (ESDs). ESDs are widely used in transportation and large-scale industries. There are various types of ESDs, such as batteries [1], fuel cells [2], and supercapacitors (SCs) [3]. Among these devices, batteries have the highest number of users due to their high energy density and large cyclability. However, their low power density allows researchers to look for alternative solutions, i.e., fuel cells that store green energy in the form of hydrogen and are currently in the developing face, which may enter the market soon. Moreover, the SCs of these ESDs have high power density and can be used where power density is required. There are various types of SCs. Based on the principle of energy storage mechanism, they are of two types: one electric double-layer capacitance (EDLCs) and pseudocapacitor [4]. The EDLC stores energy through electrostatic interaction between the ions, forming the double layer on the electrode surface [5]. The pseudocapacitor stores charges through the electrochemical faradaic redox reactions [6]. Among these two, pseudocapacitors generally have high specific capacitance (C_sp_) and low capacitance retention, whereas EDLCs have low specific capacitance compared to pseudocapacitors but have better capacitance retention power. Therefore, a hybrid type of SCs has been introduced where each mechanism takes place simultaneously [7]. Apart from all this, the synthesis route and types of materials used for developing SC play a vital role in their performance.

Scientists worldwide use different types of methods such as solvothermal [8], hydrothermal [9], chemical vapor deposition (CVD) [10], thermal deposition [11], and some green synthesis route, that affect the morphology and performance of the devices [12,13]. These chemical synthesis techniques cause toxic materials to get absorbed on the surface. Therefore, growing interest in green chemistry methods has caused the improvement of an eco-friendly approach to producing nanoparticles (NPs) [14]. The most attractive part of the green synthesis route is that it is economical, clean, non-toxic, and applied to produce inorganic NPs using various biological systems, such as plant extracts such as the leaves of many plants containing the hydroxyl (-OH) group, microorganisms, such as fungi and bacteria containing enzymes, proteins for surface phenomena and fruit extracts such as orange and lemon peels containing flavonoids [15], limonoids [16], and carotenoids [17] containing the -OH group, which act as the capping agent [18] during the synthesis of NPs. Generally, SCs have been fabricated using various types of NPs such as transition metal oxides (TMOs) [19], transition metal hydroxides (TMOHs) [20], transition metal chalcogenides (TMCs) [21], MXenes [22], metal-organic framework (MOF) [23], and zeolitic imidazolate framework-8 (ZIF-8) [24]-based NPs, etc.

Among all NPs, TMOs are better suited for developing SCs due to their unique physical, optical, and electrochemical properties. Their surface area highly depends on the material’s shape, size, and morphology. They are developed using the d-block element, partially filled with d-subshell and oxygen (O_2_) as their oxide part. The introduction of O_2_ enhances chemical properties and the surface energy of TMO-based structures. They have a high surface area and high surface area to volume ratio. TMOs have a high specific capacitance of 100–2000 F/g, high energy density when compared to carbon-based NPs, and better chemical stability than conductive polymers. Due to the presence of O_2_ in TMO, the redox reaction takes place rapidly, making them suitable materials for fast charging and discharging devices. TMOs agglomerate easily due to their rapid reaction and mass loading. Doping them with suitable conducting elements and synthesizing with suitable methods will lead to a stable rapid reaction, whereas a suitable amount of mass loading is required to achieve a high C_sp_ [25]. These shortcomings can be overcome by incorporating conductive materials such as sulfur and phosphorus into TMO-based NPs, enhancing ionic and electrical conductivity via synergism. 

Among these TMOs, cobalt oxide-based nanomaterials and different phases have gained popularity due to their non-toxicity, ease of synthesis process, long cyclic stability, and high corrosion resistance. Liu et al. [26] and group synthesized the Co_3_O_4_ nanoplates using laser ablation in liquid, demonstrating a specific capacitance of 165 F/g at 1 A/g in the KOH electrolyte. Madhu et al. [27] fabricated the nanorods-based Co_3_O_4_ composite using the hydrothermal facile synthesis route, demonstrating a specific capacitance of 94 F/g at 1 A/g. Kandalkar et al. [28] fabricated Co_3_O_4_ using the facile chemical method without any surfactant, showing the nanoplate-type morphology. The obtained Co_3_O_4_ demonstrated a specific capacitance of 118 F/g at 1 A/g with a drop-in capacitance retention of 70% after 5000 cycles. 

Here, we synthesized the TMO-based CoO and Co_3_O_4_ NP via peel extracts of oranges that were further transferred for sulfurization and phosphorization to study their effects along with the phase transformation. The comparison between the green synthesis prepared Co_x_O_y,_ and the chemically synthesized Co_x_O_y_ is also demonstrated for SC application.

## 2. Experimental Section

### 2.1. Materials and Method

Mandarin Oranges (*Citrus reticulata*) were purchased from Walmart, USA. Orange peels were collected, considered a waste product, and utilized to prepare cobalt oxide NPs. Chemicals such as sodium hypophosphite (NaPO_2_H_2_·H_2_O, Sigma Aldrich, Saint Louis, MO, USA), cobalt nitrate hexahydrate (Co(NO_3_)_2_·6H_2_O, Strem Chemicals, Newburyport, MA, USA), potassium hydroxide (KOH), sodium hydroxide (NaOH), sodium thiosulfate (Na_2_S_2_O_3_·5H_2_O), ethanol and deionized water (D.I.) were purchased from Fischer Scientific, Philadelphia, PA, USA.

### 2.2. Preparation of Peel Extracts

To make an extract, orange fruits were washed and left to dry at room temperature before being peeled. The collected peels were placed in an oven at 60 °C overnight. The fine powder was obtained after grinding a dried peel. Next, 2 g of powdered peel were dissolved in 50 mL of DI water and stirred for 3 h at room temperature. Afterward, the mixture was placed in a water bath at 60 °C for 60 min. In the end, the mixture was filtered, and the supernatant was collected to further synthesize metal oxide nanoparticles.

### 2.3. Synthesis of CoO/Co_3_O_4_ Nanoparticles

Typically, 2 g of Co(NO_3_)_2_.6H_2_O were dissolved into 42.5 mL of the as-prepared above peel extract and stirred at room temperature for 60 min. The beaker was transferred to a water bath at 60 °C for 1 h. Further, the mixture was dried overnight at 60 °C. The obtained CoO was ground into fine powder and it was characterized later. CoO is referred to as CoO-OP. To prepare the Co_3_O_4_ phase of cobalt oxide, a half of the sample was calcined at 400 °C at 5 deg./min. The nanoparticles obtained by heat treatment were labeled as Co_3_O_4_-OP-C. Hence the cobalt oxide nanoparticles were obtained using orange peels as a reducing ligand.

### 2.4. Synthesis of Cobalt Oxide Using NaOH

The Co(NO_3_)_2_·6H_2_O (2 g) was poured into 40 mL of (1N) NaOH and stirred for 60 min. The beaker was placed in a water bath at 60 °C for 1 h. The mixture was dried overnight at 60 °C. Hence Co_3_O_4_ particles were obtained without calcination and characterized. The peel extract reducing agent was replaced with NaOH to facilitate the comparative study of the phases. Hence, the obtained cobalt oxide nanoparticles using NaOH (Co_3_O_4_-NaOH) were referred to as a reference nanoparticle for the study.

### 2.5. Sulfurization

The sulfurized cobalt oxide nanoparticles were synthesized by dissolving Na_2_S_2_O_3_·5H_2_O with the already prepared CoO-OP nanoparticles, according to a ratio of 1:10 in water and transferred to the Teflon-lined autoclave (45 mL) under 160 °C for 24 h, then cooled down at room temperature. Further, the obtained material was washed several times with DI water and ethanol, placed to centrifugation for 10 min, and kept for drying in an oven at 70 °C overnight. The dried sample was taken out and ground to an exceptionally fine powder, and hence cobalt sulfide nanoparticles were formed. The Co_2_S phase was obtained from the CoO-OP and referred to as CoO-S-OP. Therefore, the same steps were used to prepare the Co_3_S_4_ phase from an annealed sample of Co_3_O_4_-OP-C, which is referred to as Co_3_O_4_-S-OP-C, and Co_3_S_4_-S-NaOH was obtained from Co_3_O_4_-NaOH.

### 2.6. Phosphorization

NaPO_2_H_2_.H_2_O in one ceramic dish and the already prepared CoO-OP NPs in another dish were taken and annealed at 350 °C in a tube furnace for 5 h under the presence of Argon (Ar) atmosphere with a heating rate of 5 °C/min. The cobalt phosphide NPs (CoP) were obtained. The CoP phase was obtained for all the samples. The CoO-P-OP, Co_3_O_4_-P-OP-C, and Co_3_O_4_-P-NaOH are referred for CoO-OP, Co_3_O_4_-OP-C, and Co_3_O_4_-NaOH nanoparticles, respectively. The schematic diagram of the synthesis of nanoparticles is provided in Figure 1 along with the proposed mechanism of the synthesis of the cobalt oxide nanoparticles by using peel extract. The presence of hydroxide (-OH) in the peel powder helps as a capping agent and as a possible reaction mechanism for the preparation of the metal oxide nanoparticles using orange peel extract in which ligation takes place between the functional components of the orange peel and cobalt precursor [29]. Organic substances such as flavonoids, limonoids, and carotenoids are present in orange peel extract, which acts as a ligand agent. One of the extract component’s hydroxyl aromatic ring groups reacts with a precursor and forms a complex-ligand with Co ions. Chemical reactions such as nucleation and shaping help in the formation of stabilized nanoparticles. Then the mixture of the organic solution is decomposed, the precipitate is calcined in the presence of air, and cobalt oxide nanoparticles are released [30].

## 3. Results and Discussion

### 3.1. Physicochemical Characterization

The successful preparation of the cobalt oxide nanoparticles by using orange peel extract and NaOH was followed by phosphorization and sulfurization. Hence, the surface morphology of the prepared samples was characterized by scanning electron microscopy (SEM) (at 10 keV). X-ray diffraction was used to study the phase purity using a Lab X (XRD-6000) Shimadzu X-ray diffractometer employing Cu K_α_ radiative instrument. To investigate insights into the structure and morphology of the particles of the synthesized materials, SEM was conducted and showed the surface structure of the materials at the scale of 200 nm. We could see the irregular sponge-like structure with the uneven pore size distributed nonuniformly throughout the CoO-OP (Figure 2a). By contrast, the sponge-like structure appears in the Co_3_O_4_-OP-C (Figure 2b) with smaller pores than the CoO-OP. However, the zig-zag continuous flakes were observed in the Co_3_O_4_-NaOH (Figure 2c) precursor. Sulfurization and phosphorization adversely affect the morphology of the material. Hence, the sponge-like morphology changed into small globular-like clouds, as is observed in Figure 2d of the CoO-S-OP. Figure 2e demonstrates the uneven open flower-like structure for the Co_3_O_4_-S-OP-C. However, a dense granular structure was observed in the Co_3_O_4_-S-NaOH (Figure 2f) and Co_3_O_4_-P-NaOH (Figure 2i). The web-like structure was observed with circular pores of uneven dimensions in the phosphorized samples of the CoO-P-OP (Figure 2g) and Co_3_O_4_-P-OP-C (Figure 2h).

The XRD patterns of the sample CoO-OP (Figure 3a) obtained before calcination showed peaks at 36.6°, 42.5°, 61.98°, 73.94°, and 77.66° to the planes of (111), (200), (220), (311) and (222) matched well with the JCPDS card No. 75-0393. The Co_3_O_4_-OP-C and Co_3_O_4_-NaOH are given in Figure 3b,c which exhibit the diffraction peaks at 19.16°, 31.64°, 37.08°, 38.46°, 45.06°, 59.4° and 65.36° to the planes of (111), (222), (311), (222), (400), (511), and (440) (JCPDS card No. 74-1656) [31]. The sample Co_3_O_4_-OP-C and Co_3_O_4_-NaOH signify a face-centered cubic structure. The obtained crystalline structure of the Co_3_O_4_ phase has a lattice constant of a = 8.076 Å and a space group of Fd3m. The Debye–Scherrer equation was used for the calculation of the average particle size: D_XRD_ = 0.9λ/(βcosθ) where D_XRD_ is the average crystalline size, λ is the wavelength of Cu-Kα, β is the full width at half maximum (FWHM) of the diffraction peak and θ is the Bragg’s angle. Furthermore, after the sulfurization process, the peaks of the CoS_2_ phase were observed for the CoO-S-OP. The main characteristic peaks and relative intensities match significantly with (210), (211), and (230) shown in Figure 3d as JCPDS No. 41-1471. The dimmer is formed by sulfur atoms covalently, and the divalent Co^2+^ cation is placed in the center of an octahedron of six S_2_^2−^ anions. Hence each sulfur is incorporated into three different octahedra and a single dimmer. Therefore, the CoO-S-OP showed an octahedral crystal structure [32].

Figure 3e,f show a typical XRD pattern of the as-prepared Co_3_O_4_-S-OP-C and Co_3_O_4_-S-NaOH samples, which represents the Co_3_S_4_ phase with the standard pattern of JCPDS No. 42-1448 at 2θ = 31.4°, 36.5°, 37.02°, 38.7°, 44.8°, and 59.4°, and corresponds to the (220), (200), (331), (222), (400), and (511) lattice planes, respectively, according to the JCPDS No. 42-1448 [33,34,35]. As the phosphorization reaction happens from the surface, the obtained products are most likely to be inhomogeneous because of the reaction surface area in the crystal architecture. The development of domains such as the CoP (Co^+^ and P^−^) phase was examined. Moreover, the rationalized phase was balanced with the valence states of the Co ions and the available amount of phosphorus (P). The product’s crystal anatomy was determined by XRD in Figure 3g–i, which incorporated the structural analysis of the phase and showed that three of the synthesized CoO-P-OP, Co_3_O_4_-P-OP-C, and Co_3_O_4_-P-NaOH materials have approximately similar atomic packaging for P. Herein, the diffraction pattern of CoP cited at 31.6°, 36.3°, 46.2°, 48.1° and 56.7° can be evaluated as the (011), (111), (112), (211), and (301) planes (JCPDS 29-0497), which showed an orthorhombic crystal geometry [36]. The CoP phase (a = 5.08 (2), b = 3.282 (11), c = 5.66 (2) Å) includes a nanocrystalline and P deficient domain. Therefore, the structure of CoP distorted to octahedral and shows an octahedral connectivity around Co.

### 3.2. Electrochemical Testing

The electrochemical measurements were carried out at room temperature using a three-electrode system using a Versa stat 4–500 electrochemical workstation (Princeton Applied Research, TN, USA). The supercapacitor performance was analyzed through cyclic voltammetry (CV) and galvanostatic charge–discharge (CD), and electrochemical impedance spectroscopy (EIS) and stability were performed to record the best outcome for the specific capacitance among the synthesized nanoparticles in a 3 M KOH solution. Electrochemical characterization was stabilized by accounting for a three-electrode system. For supercapacitor tests, a saturated mercury/mercury oxide (Hg/HgO), platinum (Pt), and the synthesized material on Ni foam were used as a reference electrode, counter electrode, and working electrode, respectively. The electrochemical performance and specific capacitance of the Ni foam loaded with the prepared NPs electrodes were investigated in the operating voltage window of 0–0.6 V.

Electrodes were prepared in the consideration of the set standard parameters for electrochemical testing. The prepared samples PVDF: NMP were taken in the ratio of 8:1:1 to form the slurry, where PVDF is poly-vinyl di-fluoride and NMP refers to n-methyl polypropylene. The already-prepared slurry was spread uniformly on the Ni foam via dipping and then transferred to the vacuum and dried at 70 °C for 48 h. The active mass loading of the best sample (CoO-P-OP) is 2.61 mg. The faradic redox curves were obtained through CV testing, which indicates the pseudocapacitive and/or EDLC behavior. EDLC indicates the adsorption or desorption mechanism during the charge storage in the pseudocapacitive behavior of the cobalt oxide nanoparticles, in addition to the excellent capacitive retention, with a rate of more than 80% from 2 to 300 mV/s. The samples explain the pseudocapacitive traits, which are confirmed by the pair of redox peaks at each scan rate. As noticed, the area of the CV curve increases with the increase in the scan rate, which justified the diffusion-controlled reaction kinetics and has two sets of redox peaks. Hence, the possible faradaic reactions for the CoO-OP can be obtained by Equations (1) and (2) [37], and those for Co_3_O_4_-OP-C and Co_3_O_4_-NaOH can be obtained with Equations (3) and (4) [38], which are arbitrated by OH^−^ ions in the alkaline medium:(1)$$\mathrm{CoO}+{\mathrm{OH}}^{-}⇌\mathrm{CoOOH}+e^{-}$$
(2)$$\mathrm{CoOOH}+{\mathrm{OH}}^{-}⇌{\mathrm{CoO}}_{2}+H_{2}O+{\;e}^{-}$$
(3)$$\mathrm{CoOOH}+{\mathrm{OH}}^{-}+H_{2}O⇌3\mathrm{CoOOH}+e^{-}\;$$
(4)$$\mathrm{CoOOH}+{\mathrm{OH}}^{-}⇌{\mathrm{CoO}}_{2}+H_{2}O+{\;e}^{-}$$

A better rate performance than the CoO-OP, Co_3_O_4_-OP-C, and Co_3_O_4_-NaOH was observed by the sulfurized nanoparticles. Therefore, the shape of the CV curve in Figure 4a–f and Appendix A is maintained by two pairs of notable redox reactions; significant peaks were noticed in the CV curves, which may occur due to the quasi-reversible [39] electron transfer processes of Co^2+^ ↔ Co^3+^ and Co^3+^ ↔ Co^4+^ within the electroactive materials. Thus, most reported cobalt sulfides show pseudocapacitive traits. The redox reaction for the CoO-S-OP is indicated in Equations (5) and (6) [40]; Co_3_O_4_-S-OP-C and Co_3_O_4_-S-NaOH are indicated in Equations (7) and (8) [41].
(5)$${\mathrm{CoS}}_{2}+{\mathrm{OH}}^{-}⇌{\;\mathrm{CoS}}_{2}\mathrm{OH}+H_{2}O+e^{-}$$
(6)$${\mathrm{CoS}}_{2}\mathrm{OH}+{\mathrm{OH}}^{-}⇌{\;\mathrm{CoS}}_{2}O+H_{2}O+e^{-}$$
(7)$${\mathrm{Co}}_{3}S_{4}+{\mathrm{OH}}^{-}⇌{\;\mathrm{Co}}_{3}S_{4}\mathrm{OH}+e^{-}\;$$
(8)$${\mathrm{Co}}_{3}S_{4}\mathrm{OH}+{\mathrm{OH}}^{-}⇌{\;\mathrm{Co}}_{3}S_{4}O+H_{2}O+e^{-}$$

The proposed redox reactions for the CoO-P-OP, Co_3_O_4_-P-OP-C, and Co_3_O_4_-P-NaOH are indicated in Equations (9)–(12) [42], which are based on the CV curve. The CoO-P-OP, Co_3_O_4_-P-OP-C, and Co_3_O_4_-P-NaOH showed the same CoP phase in XRD (Figure 3g–i). According to the already reported results on the CoP phase, the material for supercapacitor exhibits the overall redox reaction (Equations (9)–(12)), which is caused by the Co(OH)_2_ that is obtained due to oxidation of CoP phase at low voltage and the Co_x_P_y_O_z_ surface formed due to chemical reaction. Furthermore, it is recorded that CoP exhibits comparable much higher current densities than other samples in the literature for phosphorized cobalt oxide. Element P was lost after the stability test and P might transfer into PO_4_^3−^ and mixed in water [43].
(9)$$\mathrm{CoP}+2{\mathrm{OH}}^{-}⇌\;\mathrm{Co}{\left(\mathrm{OH}\right)}_{2}+P+2e^{-}$$
(10)$$\mathrm{Co}{\left(\mathrm{OH}\right)}_{2}+{\mathrm{OH}}^{-}⇌\;\mathrm{CoOOH}+H_{2}O+2e^{-}\;$$
(11)$$\mathrm{CoOOH}+{\mathrm{OH}}^{-}⇌{\;\mathrm{CoO}}_{2}+H_{2}O+e^{-}$$
(12)$${\mathrm{Co}}_{x}P_{y}O_{z}+{\mathrm{OH}}^{-}⇌{\;\mathrm{Co}}_{x}P_{y}O_{z}\mathrm{OH}+e^{-}$$

Moreover, there is no drastic change in the shape of the CV curves at various current densities, which indicates the best reversibility with enhanced mass transportation. The phosphorized samples have a larger area under the CV curves, which indicates a higher ability to store charges. Furthermore, as noticed, the area of the CV curve increases with the increase in the scan rate, which justified the diffusion-controlled reaction kinetics and holds two sets of redox peaks. The following equation was used to study the charge storage mechanism (Equation (13)) [44].
(13)$$i=av^{b}$$
where i is the measure of peak current, *v* is scan rate, and *a* and *b* are the parameters. The charge storage mechanism is influenced by the diffusion-controlled faradic reaction if *b* is 0.5, and when *b* is 1 the total charge performance is affected by the capacitive surface mechanism. Figure 4g displays the slope of the three best samples out of the nine synthesized samples; the CoO-OP, CoO-S-OP, and CoO-P-OP electrodes’ *b* value is 0.83, 0.71, and 0.81, respectively. These obtained values indicate that the electrodes have a hybrid charge storage mechanism and it was observed in detail by using the following equation (Equation (14)) [44], in which $k_{1}v$ is the capacitive effect and $k_{2}v^{1/2}$ is the diffusion-controlled effects.
(14)$$i=k_{1}v+k_{2}v^{1/2}\;$$

In Figure 4h,i, both the traits were analyzed at 5 mV/s and 100 mV/s scan rates. At a low scan rate (5 mV/s), the diffusion and capacitance contributions were 77% and 23%, 87% and 13%, and 82% and 18%, respectively. However, at a high scan rate (100 mV/s), these contributions were 62% and 38%, 61% and 39%, and 50% and 50% for the CoO-OP, CoO-S-OP, and CoO-P-OP electrodes, respectively. Higher specific capacitance is due to a diffusion-controlled reaction in which the electrolyte penetrates the deep nanomaterial interlayers. On the other hand, at a higher scan rate, the electrodes possessed a lower specific capacitance, which may be because of the surface phenomenon. SEM of the CoO-P-OP (Figure 2g) showed a more porous structure with the thin wires which gives the shape of a web, and this increases the active surface sites for the ions to penetrate inside the material. Furthermore, the electrochemical performance of the CoO-P-OP is better than the CoO-S-OP and CoO-OP, which can be reasoned to the lower electronegativity of P, and the ability to fit in the interstitial position between the metal atoms. Due to its optimum size, it behaves as a site of a positive radical and is rich in holding and entangling positive species [45]. However, the associated metal behaves as an electropositive and traps the lone pair species or electrons. Therefore, the introduction of P in a cobalt oxide nanoparticle can probably enhance its electrochemical activity [46].

All the samples showed good symmetry and high coulombic efficiency because of the diffusion-controlled phenomenon during the faradic electron transfer with a fast reaction rate and less time for the electrons to couple; and due to polarization, the reduction peaks shift swiftly to the left, which resembled the high scan rate [47]. On the other hand, the oxidation peaks shift toward the right. The high scan rate of the current is recorded for the phosphorized samples among all because the electrons took less time to emit and absorb, which resulted in the peak shifting and fast scan rate. Moreover, the highest surface area noticed in the phosphorized electrodes of the CV curves indicates their ability to store a greater amount of charge for the higher specific capacitance. The above fact is further confirmed by the GCD curves (Figure 5a–i), where the discharge time is related to a specific capacitance, which means that the longer the discharge time, the higher the specific capacitance. The relation is given below (Equation (15)) [48,49]:(15)$$C=\frac{I\Delta t}{m\Delta V}$$

Herein, C is the specific capacitance (F/g), I is the current (A), Δt is the discharge time (s), m is the mass of the active material (g) and ΔV is the potential window.

The GCD for all the samples are performed at various current density ranging from 1 A/g to 30 A/g in the potential range from 0 to 0.6 V. Multiple charge and discharge experiments were carried out. Among all the samples, the noticeable specific capacitance was recorded for the CoO-P-OP as 185 F/g at 1 A/g. The charge–discharge curve delineates that the material is stable up to 5000 charge and discharge cycles. All the synthesized materials showed stable charge–discharge properties, and the nature was asymmetric. Zhou et. al. [43] studied the electrochemical study of the phosphorized materials and observed a couple of distinct faradaic peaks that define the existence of redox behavior in the electrochemical process because of the conversion of Ni^(δ+)^/Co^(δ+)^ into the Ni^(2+)^/Co^(2+)^ state in NiCoP/C, Ni_2_P/C and Co_2_P/C. Moreover, the small separation between the anodic and cathodic peaks was observed and indicated the good electrochemical reversibility of the composites. Therefore, the GCD curve has a consistent potential with the CV curves due to polarization, and hence they provide an enhanced specific capacitance. Similarly, the change of the valence state might cause the phosphorized sample to produce effective results for higher capacitance in this study too.

In addition, to examine the interfacial electrochemical behaviors of the Ni foam-based electrodes, EIS was measured, which incorporates the information about the resistance offered by the material during the charge transfer. The EIS test was carried out to observe the Nyquist plots from which the impedance data were analyzed and showed a semicircle in the high-frequency range followed by a straight line in the low-frequency range, designating that the electrochemical process on the surface of the material of the electrode is kinetically diffusion administered. The curve can be disintegrated into three parts. The intersection with the x-coordinate in the high-frequency range refers to the electrolyte resistance (R_s_), which defines the ionic resistance of the electrolyte, the intrinsic resistance of the electrode, and the interface resistance. The charge transfer resistance (R_ct_) is deduced from the diameter of the semicircle. The plots are shown in Figure 6a, and the values of R_s_ and R_ct_ for the fabricated electrodes are shown in Table 1.

The CoO-P-OP and Co_3_O_4_-P-OP-C displayed the approximately similar and smallest R_s_ value detecting the high conductivity. However, the CoO-P-OP showed the least value of R_ct_ of 1.52 Ω among all the synthesized samples, indicating a low-charge transfer hindrance and promoting more movement of electrons. Therefore, the highest specific capacitance is noticed for CoP 185 F/g at 1 A/g because of the doping of P in the already prepared CONPs, which is derived by using peel extract, and the low C_sp_ of the CoO-OP, Co_3_O_4_-OP-C, and Co_3_O_4_-NaOH may be attributed to its structure loss in the electrolytic system during the electrochemical study. The specific capacitance is illustrated in Figure 6b. Moreover, it improved its energy and power density (Figure 6c). The energy density and power density are derived from Equations (16) and (17) [50,51,52]:(16)$$E=\frac{1\;\left(C\Delta V^{2}\right)}{2}$$
(17)$$P=\frac{E}{\Delta t}$$

The energy density of the CoO-P-OP can approach about 9 Wh/kg at a power density of 296 W/kg. The comparison data between C_sp_, the power density, and the energy density of all the synthesized samples are illustrated in Appendix A. In addition, the cycling performance was investigated at a current density of 10 A/g for 5000 cycles, the retention capacity of all the samples maintaining at around 85%. Figure 6d shows the retention capacity of the CoO-P-OP, which has the highest specific capacitance with the 5 cycles at the beginning and end to show the consistency of the charge–discharge effect. Better electrical conductivity results in low resistivity, which implies better change transfer and hence enhances the capacitance of a material. Therefore, the CoO-P-OP showed improved results among the other synthesized samples. Table 2 shows the comparison between our sample and some other work.

## 4. Conclusions

To recapitulate, this work deals with the green route of synthesis, and the further produced NPs are followed by phosphorization and sulfurization to test the comparative study of the phases and the electrochemistry. Orange peel is a bio-waste material, and the waste was utilized in this study as a reducing agent. Co-nanoparticles were produced by the precipitation method. The crystalline quality with the well-defined peaks and structural history was recorded by XRD. All samples were tested for supercapacitor study and all phosphorized samples showed better electrical conductivity, which results in a bountiful specific capacitance because of the low EIS, a high area in the CV curve, and the high discharge time, recorded in the GCD plots. The synthesized nanoparticles, by using plants, showed the best results and observed a good columbic efficiency, energy, and power density. The stability of the materiassl is consistent and impactful by 85% and the cycles from the beginning and end follow the same trend, hence proving that the nanoparticles derived by using biomaterial are more stable and sustainable.

There is an enormous scope for improving the capacitive property of the metal oxide nanoparticles by using plant extracts such as the incorporation of carbon quantum dots in making composites, reduced graphene, MXenes, phosphides and sulfides for the improvement of the supercapacitive attributes, and research in this direction would evolve a plethora of significant materials for supercapacitors.

## Figures and Tables

**Figure 1 nanomaterials-12-04119-f001:**
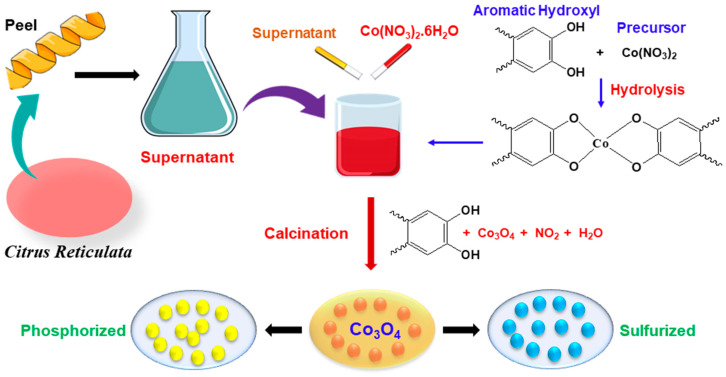
Schematic diagram of the synthesis of nanoparticles, along with the chemical mechanism of Co_3_O_4_ nanoparticle formation.

**Figure 2 nanomaterials-12-04119-f002:**
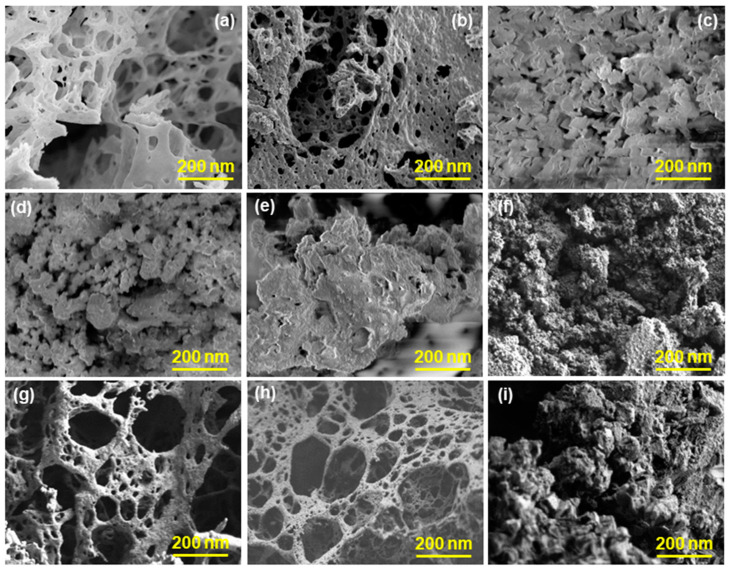
The SEM images of the prepared (**a**) CoO-OP, (**b**) Co_3_O_4_-OP-C, (**c**) Co_3_O_4_-NaOH, (**d**) Co_3_O_4_-S-OP, (**e**) Co_3_O_4_-S-OP-C, (**f**) Co_3_O_4_-S-NaOH, (**g**) CoO-P-OP, (**h**) Co_3_O_4_-P-OP-C, (**i**) Co_3_O_4_-P-NaOH nanomaterials at the scale of 200 nm.

**Figure 3 nanomaterials-12-04119-f003:**
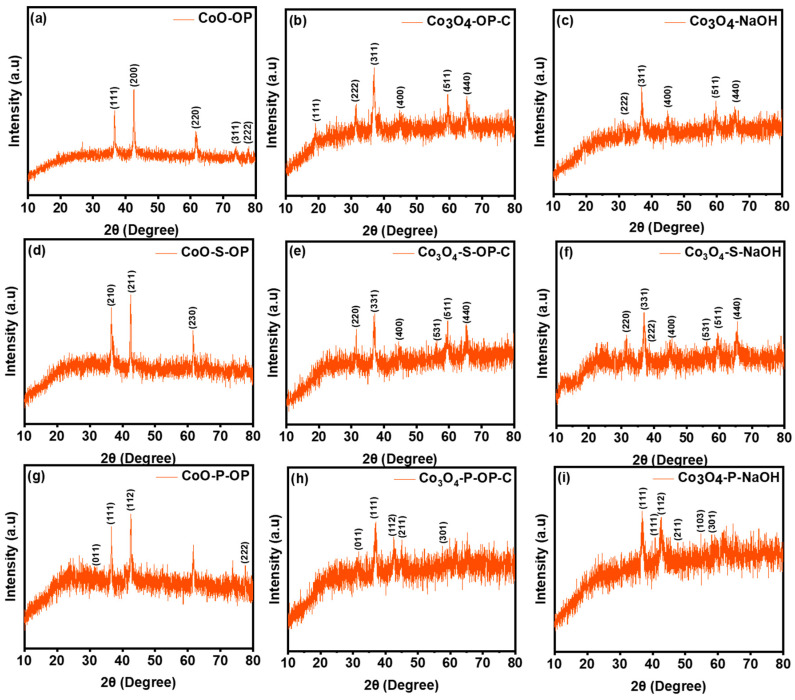
X-ray diffraction patterns of the prepared (**a**) CoO-OP, (**b**) Co_3_O_4_-OP-C, (**c**) Co_3_O_4_-NaOH, (**d**) Co_3_O_4_-S-OP, (**e**) Co_3_O_4_-S-OP-C, (**f**) Co_3_O_4_-S-NaOH, (**g**) CoO-P-OP, (**h**) Co_3_O_4_-P-OP-C, (**i**) Co_3_O_4_-P-NaOH, nanoparticles.

**Figure 4 nanomaterials-12-04119-f004:**
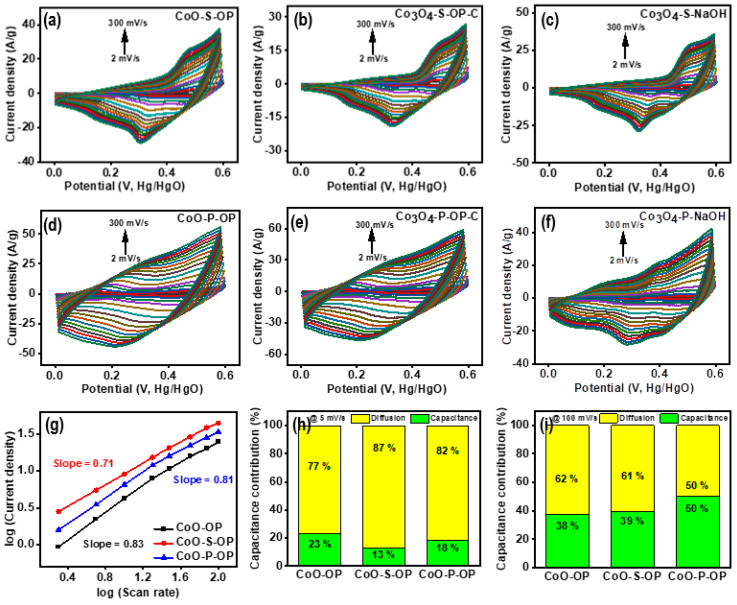
The CV scan curve from 2 to 300 mV/s for prepared (**a**) Co_3_O_4_-S-OP, (**b**) Co_3_O_4_-S-OP-C, (**c**) Co_3_O_4_-S-NaOH, (**d**) CoO-P-OP, (**e**) Co_3_O_4_-P-OP-C, (**f**) Co_3_O_4_-P-NaOH nanoparticles, and (**g**) log (current) versus log (scan rate) plot, (**h**,**i**) diffusion and capacitance contribution for CoO-OP, CoO-S-OP, and CoO-P-OP at a scan rate of 5 mV/s and 100 mV/s, respectively.

**Figure 5 nanomaterials-12-04119-f005:**
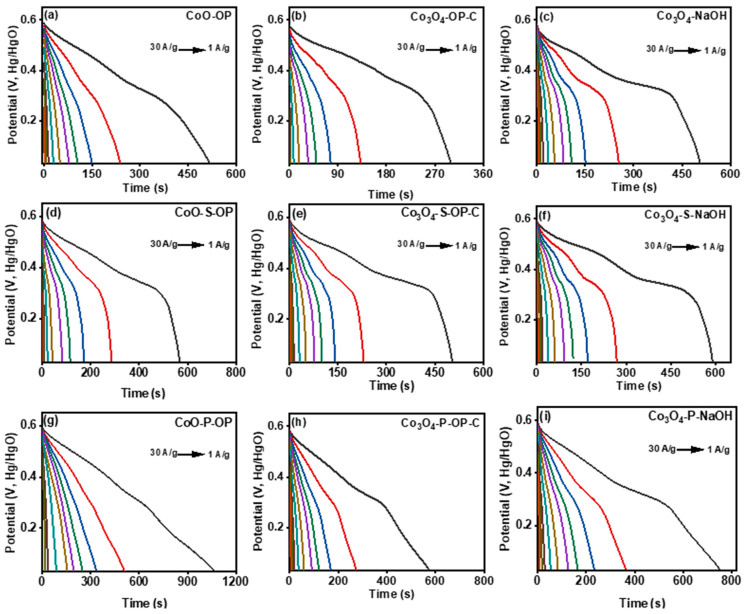
The discharge curves of the prepared (**a**) CoO-OP, (**b**) Co_3_O_4_-OP-C, (**c**) Co_3_O_4_-NaOH, (**d**) Co_3_O_4_-S-OP, (**e**) Co_3_O_4_-S-OP-C, (**f**) Co_3_O_4_-S-NaOH, (**g**) CoO-P-OP, (**h**) Co_3_O_4_-P-OP-C, (**i**) Co_3_O_4_-P-NaOH, nanoparticles, from current density 30 A/g to 0.5 A/g.

**Figure 6 nanomaterials-12-04119-f006:**
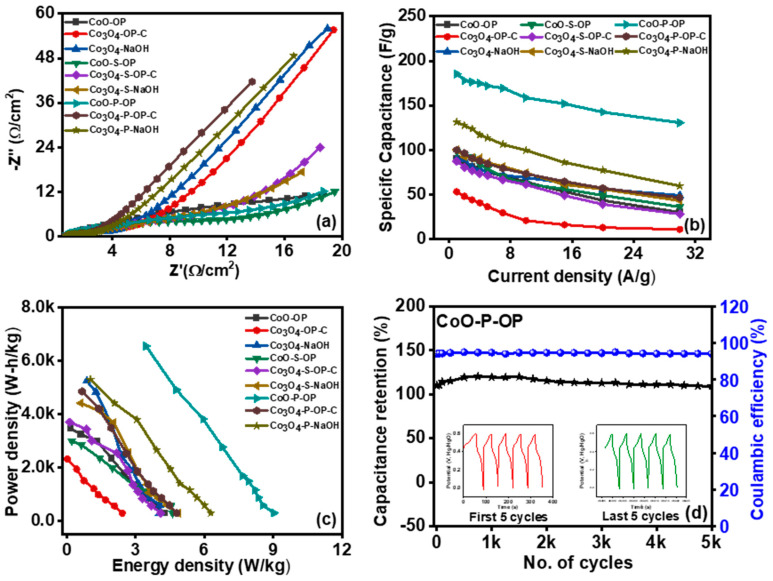
(**a**) EIS, (**b**) specific capacitance vs. current density, (**c**) power density vs. energy density, of all the prepared samples (**d**) Stability curve for the CoO-P-OP sample.

**Table 1 nanomaterials-12-04119-t001:** The calculated values of Rs and R_ct_ for the fabricated electrodes.

Sample Name	CoO-OP	Co_3_O_4_-OP-C	Co_3_O_4_-NaOH	CoO-S-OP	Co_3_O_4_-S-OP-C	Co_3_O_4_-S-NaOH	CoO-P-OP	Co_3_O_4_-P-OP-C	Co_3_O_4_-P-NaOH
R_s_ (Ω)	0.95	0.75	0.99	1.32	1.26	1.59	0.70	0.705	0.76
R_ct_ (Ω)	3.85	3.25	3.71	6.19	4.24	4.73	1.52	2.81	2.42

**Table 2 nanomaterials-12-04119-t002:** The comparison of specific capacitance, energy density, and power density of our samples with other reported work.

Sample	Specific Capacitance F/g @ 1A/g	Energy Density Wh/kg	Power Density W/kg	Ref.
Ultrafine Co_3_O_4_	165	3	31	[26]
C/Co_3_O_4_-750	44	1.66	18.75	[53]
CFAC-800	133	-	-	[54]
AC/Co_3_O_4_-NP	182	25.27	585	[55]
CoMoO_4_	128	7.3	146	[56]
Co_3_O_4_/r-GO	163.8	-	-	[57]
CoO/CNT	17.4 @ 0.25 A/g	3.48 mWh/g @ 0.25 A/g	-	[58]
N,B, Co doped C	184	18.7	400	[59]
Mesoporous C/Co	113	-	-	[60]
Needle-like cobalt oxide/graphene composite	60	-	-	[61]
Pongam seed shell derived Carbon and Co_3_O_4_	94	-	-	[27]
Co_3_O_4_ NPs	120	-	-	[62]
Co_3_O_4_	118	5.8	330	[28]
CoO-OP	90	4.2	292	Our work
CoO-S-OP	98	4.6	291	Our work
CoO-P-OP	185	9	296	Our work

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
