# Peer review of "Waste Citrus reticulata Assisted Preparation of Cobalt Oxide Nanoparticles for Supercapacitors"

_nanomaterials, 2022, doi:10.3390/nano12234119_

Round 1
Reviewer 1 Report
This work deals with the green route of synthesis, and further produced NPs of cobalt are followed by phosphorization and sulfurization to test the comparative study of phases and electrochemistry. The cobalt oxide nanoparticle showed a specific capacitance of 90 F/g at 1 A/g, whereas the cobalt sulfide and phosphide samples delivered an improved specific capacitance of 98 F/g and 185 F/g at 1 A/g. This topic is interesting and conclusion are well supported by the results and comprehensive characterizations. I suggest it for publication after addressing the following problems.
Comment 1: XRD datas in Figure 3 are too coarse for publication.
Comment 2: what’s the mass loadings of active materials in the electrode. This parameter is critically important for fairly evaluating their electrochemical performance.
Comment 3: The specific capacitance (199~200 F/g) is not high compared to other Pseudocapacitive nanomaterials (300~600 F/g) with electrochemical faradaic redox reactions. Authors are suggested to comment on this.
Comment 4: I wonder that the gravimetric pseudocapacitance is normalized by the mass of active materials or whole electrode materials. In the practical applications, the specific capacitance or energy/power density is normalized by the whole electrode materials. Recent study has proposed a new concept of more from less to address this issue (Journal of Power Sources 492 (2021) 229639). Authors are suggested to comment on this in the introduction.
Comment 5: From the EIS and CV also the Ragone plot, the electrical conductivity of samples is not very good. Authors should provide the size distribution of nanoparticle to analyze the poor conductivity. In this regard, superior-conductivity nanomaterials, e.g., MXene (Energies 2022, 15(3), 1191; ACS Energy Lett. 2020, 5, 7, 2266–2274), is preferable as a future perspective study in the conclusion part is needed.
Author Response
Comment 1: XRD datas in Figure 3 are too coarse for publication.
Authors comments: Thank you for your comments. The data is little course due to nanostructured nature of the materials. We have slight smooth the data without affecting the original nature of the data.
Comment 2: what’s the mass loadings of active materials in the electrode. This parameter is critically important for fairly evaluating their electrochemical performance.
Authors comments: Thank you for noticing this. Sorry for missing this important parameter. We have added the mass loading of active material in manuscript. The active mass loading of the samples was about 1.5 mg/cm2.
Comment 3: The specific capacitance (199~200 F/g) is not high compared to other Pseudocapacitive nanomaterials (300~600 F/g) with electrochemical faradaic redox reactions.
Authors are suggested to comment on this.
Authors comments: We kindly accept this comment and wanted to throw some light on the behavior and nature of plant-derived nanoparticles. The aim of this research is to study the effect of plant-derived nanoparticles and their challenges. We accept the specific capacitance was low. However, we compared our results with other works and included a few more comparison data in Table 2 to show that the synthesized cobalt oxide nanoparticles by using orange peels have better results.
Comment 4: I wonder that the gravimetric pseudocapacitance is normalized by the mass of active materials or whole electrode materials. In the practical applications, the specific capacitance or energy/power density is normalized by the whole electrode materials. Recent study has proposed a new concept of more from less to address this issue (Journal of Power Sources 492 (2021) 229639). Authors are suggested to comment on this in the introduction.
Authors comments: Thank you, sir, for your valuable suggestion. I have gone through the reference you attached Journal of Power Source 492 (2021) 22639. This paper has been written so nicely that the concept of mass loading is clear in the introduction part itself. Due to the presence of O2 in TMO, the redox reaction takes place rapidly, making them suitable materials for fast charging and discharging Devices. TMOs agglomerate easily due to their, rapid reaction and mass loading. Doping them with suitable conducting elements and synthesizing with suitable method will lead to this stable rapid reaction whereas a suitable amount of mass loading is required to achieve a high Csp.
Comment 5: From the EIS and CV also the Ragone plot, the electrical conductivity of samples is not very good. Authors should provide the size distribution of nanoparticle to analyze the poor conductivity. In this regard, superior-conductivity nanomaterials, e.g., MXene (Energies 2022, 15(3), 1191; ACS Energy Lett. 2020, 5, 7, 2266–2274), is preferable as a future perspective study in the conclusion part is needed.
Authors comments: Thank you for your thorough investigation. We have tried to explain the nature of the working of materials through EIS, CV, and Ragone plot. Moreover, we added one more table in the supplementary information (Table S1) which speaks about the data of specific capacitance and Ragone plot. In addition, we added the charge transfer resistance value in Table 1, which indicates the proficiency of the flow of charges. Moreover, we added the future perspectives in the conclusion part. Apart from this, we use the references you mentioned and it helped us a lot.
Reviewer 2 Report
In this work, an orange peel extract is used to make cobalt oxide nanoparticles from cobalt nitrate hexahydrate. Additionally, the same nanoparticles were also obtained from various precursors for phase and electrochemical behavior comparisons. Subsequently, the prepared Co-nanoparticles were also sulfurized and phosphorized. The cobalt oxide nanoparticle showed a specific capacitance of 90 F/g at 1 A/g, whereas the cobalt sulfide and phosphide samples delivered an improved specific capacitance of 98 F/g and 185 F/g at 1 A/g. The phosphide-based nanoparticles offer more than 85% capacitance retention after 5000 cycles. Overall, this work occpuies some novelty. But before its publication, some issues need to be solved. 1. Some important characterizations need to be added to observe the sulfurization and phosphorization like FTIR, Raman and XPS. 2. It seems from Fig. 5 that the Co3O4-P possesses the maximum specific capacitance, please the reason. 3. The Fig.6 shows the Co-P-OP kept the best stability. whether the electrochemical reaction process is diffent? 4. The reaction kinetics needs to be further diccussed by referring and citing related literatures: Advanced Functional Materials 2015 25 (47), 7381-7391; Electrochimica Acta 2015, 173, 399-407; Journal of Materials Chemistry A 2016 4 (48), 19026-19036. 5. There are some grammar and format issues. The authors are suggested to polish the manuscript to decrease redundancy and correct the grammar mistakes.
Author Response
Thank you for your comments. We have revised the manuscript based on your suggestions.
Reviewer 3 Report
A manuscript entitled “Waste Citrus reticulata Assisted Preparation of Cobalt Oxide Nanoparticles for Supercapacitors” can be reconsider for publication in the journal “Nanomaterials” after following major revisions.
1. The manuscript should be strongly corrected with better sentence structures and grammar. Ambiguous wording and/or grammatical/technical errors such as “..developing face..”, “..hand-mortar pistil..”, “..a half-sample..” “..illustrated below (mention figure/scheme number)..” are present throughout the manuscript.
2. The manuscript severely lacks proper referencing in numerous places, such as an insufficient reference for different energy storage devices and their electrochemical properties part, no references for the advantages of green synthesis methods and several synthesis techniques of that, not a single referencing for TMOs NPs part, surprisingly no referencing for several mathematical equations as well. This indicates a poor literature review.
3. In the Introduction, as well as the physical/electrochemical characterization part, some recent references regarding supercapacitor work such as “ACS Appl. Nano Mater. 2022, 5, 160−175”, “Chemosphere 303 (2022) 135290” etc can be mentioned.
4. The introduction section lacks satisfactory justification as to why this research is better/different when compared to the reported ones.
5. Consistency should be followed in compound nomenclatures such as either Co3O4 or cobalt oxide.
6. On page 8, line 145, texts seem to be overlapped. It is advised to examine the manuscript before submission thoroughly.
7. The physical characterization section should be strongly modified to support electrochemical results as to why certain materials showed the best electrochemical results. The data and the other characterization can be compared with reference from the (Journal of Electroanalytical Chemistry, Volume 856, 1 January 2020, 113670) with proper citation.
8. Furthermore, more characterization techniques should be performed to analyze/differentiate and justify the superiority of one sample over other. At least TEM/ HRTEM, and XPS should be performed.
9. Page 14 line 239 to line 244 discuss the contents of equations 9-12, however, equation 7-10 is referenced in line 240.
10. Page 15, lines 244 to line 252 have repetitive texts. It is advised to thoroughly examine the manuscript before submission.
11. Consistency should be maintained while reporting the numerical values such as capacitance of CoO-P-OP as 185.3 F/g at 1 A/g or 185 F/g at 1 A/g.
12. Consistency should be maintained while reporting the electrode name such as “CoP 185 F/g at 1 A/g” or “CoO-P-OP as 185 F/g at 1 A/g”.
13. Data presented in Fig. (b), and (c) should be summarized in the table and added to the ESI file.
14. Table 2 is incomplete, at least 10 more similar type of materials should be compared.
15. There is a strong belief in the supercapacitor research community that the battery type material used in supercapacitor work should strictly report the results in terms of capacity such as C/g, mAh/g, mAh/cm2, or mAh/cm3 rather than capacitance such as F/g or F/cm2 or F/cm3. Hence, it is advisable to report the results in terms of capacity as well. Details of calculation methods can be referenced from the Chemical Engineering Journal, Volume 450, Part 4, 15 December 2022, 138363; Journal of Energy Storage, Volume 33, January 2021, 102080 and Inorganics 2022, 10(6), 86 with the proper citation in this article.
16. The comparison table of several electrochemical parameters from this work with reported results seems too small to declare this work a better one. Several works need to be compared through a proper literature review.
Author Response
A manuscript entitled “Waste Citrus reticulata Assisted Preparation of Cobalt Oxide Nanoparticles for Supercapacitors” can be reconsider for publication in the journal “Nanomaterials” after following major revisions.
- The manuscript should be strongly corrected with better sentence structures and grammar. Ambiguous wording and/or grammatical/technical errors such as “..developing face..”, “..hand-mortar pistil..”, “..a half-sample..” “..illustrated below (mention figure/scheme number)..” are present throughout the manuscript.
Authors comments: Thank you for your thorough investigation. We checked the grammar and sentence structures. We did the necessary sentence transformation and structural modification.
- The manuscript severely lacks proper referencing in numerous places, such as an insufficient reference for different energy storage devices and their electrochemical properties part, no references for the advantages of green synthesis methods and several synthesis techniques of that, not a single referencing for TMOs NPs part, surprisingly no referencing for several mathematical equations as well. This indicates a poor literature review.
Authors comments: Thank you for your valuable suggestions. We have gone through the manuscript, and we accept there is a need for some valuable references. We have added some references for TMOs, and NPs (reference numbers 14, 19, 20, 21, 22, 23, 24, and 25). Also, we have added references for mathematical equations (reference numbers 44, 48, 49, 50, and 51).
- In the Introduction, as well as the physical/electrochemical characterization part, some recent references regarding supercapacitor work such as “ACS Appl. Nano Mater. 2022, 5, 160−175”, “Chemosphere 303 (2022) 135290” etc can be mentioned.
Authors comments: We cited these articles in the introduction and results and discussion.
- The introduction section lacks satisfactory justification as to why this research is better/different when compared to the reported ones.
Authors comments: Your suggestion is valuable to us. We have added some statements to support your comment. The green synthesis route for the development of TMOs allows the fabrication of crystalline NPs of cobalt oxide.
- Consistency should be followed in compound nomenclatures such as either Co3O4 or cobalt oxide.
Authors comments: We corrected and maintained consistency all over the manuscript.
- On page 8, line 145, texts seem to be overlapped. It is advised to examine the manuscript before submission thoroughly.
Authors comments: We successfully removed the overlapping and repetition of the texts and numeric.
- The physical characterization section should be strongly modified to support electrochemical results as to why certain materials showed the best electrochemical results. The data and the other characterization can be compared with reference from the (Journal of Electroanalytical Chemistry, Volume 856, 1 January 2020, 113670) with proper citation.
Authors comments: Thank you for your esteem suggestion. We added this reference in line 59. The material synthesized in Journal of Electroanalytical Chemistry, Volume 856, 1 January 2020, 113670 is chemically derived nanomaterial. However, we are synthesizing the nanoparticles by using plant waste.
- Furthermore, more characterization techniques should be performed to analyze/differentiate and justify the superiority of one sample over other. At least TEM/ HRTEM, and XPS should be performed.
Authors comments: Thank you for your esteem suggestion, however, we do not have TEM/HRTEM and XPS facility at our research center. These are additional characterization methods that will not influence the reactivity and productivity of the material, so we try to report the capacitive nature of the material through several electrochemical testing. XRD and SEM were accompanied to study the phase and morphology. It helps to understand whether the obtained sample has the desired phase or not.
- Page 14 line 239 to line 244 discuss the contents of equations 9-12, however, equation 7-10 is referenced in line 240.
Authors comments: We recorrected it and mentioned the correct equation numbers with references.
- Page 15, lines 244 to line 252 have repetitive texts. It is advised to thoroughly examine the manuscript before submission.
Authors comments: Thank you for your supervision, we have examined the manuscript now and we resolved the repetitive texts.
- Consistency should be maintained while reporting the numerical values such as capacitance of CoO-P-OP as 185.3 F/g at 1 A/g or 185 F/g at 1 A/g.
Authors comments: We totally understand the significance of consistency and checked the numerical values and words throughout the manuscript.
- Consistency should be maintained while reporting the electrode name such as “CoP 185 F/g at 1 A/g” or “CoO-P-OP as 185 F/g at 1 A/g”.
Authors comments: We rectified this mistake everywhere.
- Data presented in Fig. (b), and (c) should be summarized in the table and added to the ESI file.
Authors comments: It is a very kind suggestion, and we summarized the specific capacitance and Ragone plot in Table S1 (Supporting Information).
- Table 2 is incomplete, at least 10 more similar type of materials should be compared.
Authors comments: Thank you for your suggestion and accordingly we added a few more data in comparison table.
- There is a strong belief in the supercapacitor research community that the battery type material used in supercapacitor work should strictly report the results in terms of capacity such as C/g, mAh/g, mAh/cm2, or mAh/cm3 rather than capacitance such as F/g or F/cm2 or F/cm3. Hence, it is advisable to report the results in terms of capacity as well. Details of calculation methods can be referenced from the Chemical Engineering Journal, Volume 450, Part 4, 15 December 2022, 138363; Journal of Energy Storage, Volume 33, January 2021, 102080 and Inorganics 2022, 10(6), 86 with the proper citation in this article.
Authors comments: Thank you for your precious comment, we understand the assiduousness of the reporting units. Since, this work deals with capacitance so we reported specific capacitance in F/g. We added all the prescribed journals.
- The comparison table of several electrochemical parameters from this work with reported results seems too small to declare this work a better one. Several works need to be compared through a proper literature review.
Authors comments: We added more reported work, and the comparison is maintained accordingly.
Round 2
Reviewer 1 Report
Authors have addressed the comments well. I suggest the manuscript for publication in current version.
Reviewer 2 Report
accepted
Reviewer 3 Report
All the comments are answered properly.